

# Technical Note: A modified formulation of dynamic energy budget theory for faster computation of biological growth

Jinyun Tang and William J. Riley

Climate and Ecosystem Sciences Division, Lawrence Berkeley National Laboratory, Berkeley, CA, USA.

*Correspondence to*: Jinyun Tang (jinyuntang@lbl.gov)

Running title: the mDEB model.

**Abstract.** The mass conservation equation in the presence of boundary fluxes and chemical reactions from non-equilibrium thermodynamics is used to derive a modified dynamic energy budget (mDEB) model. Compared to the standard
dynamic energy budget (sDEB) model (Kooijman, 2009), this modified formulation does not place the dilution effect in the mobilization kinetics of reserve biomass, and it maintains the partition principle for reserve mobilization dynamics for both linear and non-linear kinetics. Overall, the mDEB model shares most features with the sDEB model. However, for biological growth that requires multiple nutrients, the mDEB model is computationally much more efficient by not requiring numerical iterations for obtaining the specific growth rate. In an example of modelling the growth of *Thalassiosira weissfloggi* in a
nitrogen-limiting chemostat, the mDEB model was found to have almost the same accuracy as the sDEB model, while requiring almost half of the computing time of the sDEB model. Since the sDEB model has been successfully applied in numerous studies, we believe that the mDEB model can help improve the modelling of biological growth and the associated ecosystem processes in various contexts.

## 1 Introduction

By aid of membranes (and cell walls), biological cells create an intracellular environment where substrates taken up from the environment are concentrated and converted into new biomass and new cells (Lodish et al., 1999). The similarity between the role played by the (membrane-confined) intracellular environment and the (container-held) aqueous solution that supports chemical reaction experiments (in the lab) has motivated the development of variable-internal-store models to model biological growth ( Nev and Van Den Berg, 2017; Grover, 1991; Droop, 1974; Williams, 1967). Among the many
formulations, the dynamic energy budget (DEB) model has been a very successful example (Kooijman, 2009; Tolla et al., 2007; Sousa et al., 2010; Matyja and Lech, 2024; Tang and Riley, 2015).

The key idea of variable-internal-store models is to represent a biological organism with one structural compartment, which holds one or multiple storage compartments (Nev and Van Den Berg, 2017; Kooijman, 2009). The production of new cells is modelled as the growth of structural biomass, as driven by the transformation dynamics of storage compartments.



These models vary in their formulation of storage dynamics, and how the turnover of storage drives structural growth. In the DEB framework, storage is termed "reserve", whereas in the Droop model, storage is termed "quota". Since here we are proposing a modified DEB model (mDEB), we use the term "reserve" hereafter. Moreover, in the DEB textbook (Kooijman, 2009), we note that standard DEB model refers to the simplest non-degenerated DEB model. In this study, we use sDEB simply to semantically contrast the formulation in Kooijman (2009) with the mDEB proposed here.

The sDEB model adopts the following three key assumptions: (1) The strong homeostasis assumption that is abstracted from observations that almost every biological organism is made up by several groups of macromolecules, e.g., carbohydrates, proteins, lipids, and nucleic acids (Lodish et al., 1999). This assumption states that the chemical composition of reserve(s) and structure(s) are constant, but their amounts can vary. (2) The weak homeostasis that is abstracted from observed stable whole organism elemental stoichiometry, such as the Redfield ratio (Redfield, 1934), which is the foundation
for the application of ecological stoichiometry theory (Sterner and Elser, 2002). This assumption states that if food density does not change, then reserve density, defined as the ratio between the amounts of reserve and structure, becomes constant while growth continues. That is, reserve and structure grow in harmony and the chemical composition of biomass does not change. (3) The principle of merging and partitioning of reserves, which is abstracted from the evolution from unicellular to multicellular life forms. These assumptions combined with Euler's theorem on homogeneous function are used to infer that
the reserve turnover dynamics must be a linear function of reserve density (see chapter 2 of Kooijman (2009) for details).

    Recently, Tang and Riley (2023) suggested that the linear reserve dynamics in the sDEB model can be replaced with a more generic nonlinear enzyme kinetics (derived from law of mass action) without violating the principle of merging and partitioning of reserves, so that a closer link can be built between the DEB model and models that consider intracellular biochemistry using explicit enzyme kinetics (Etienne et al., 2020; Tadmor and Tlusty, 2008). Tang and Riley (2023)
demonstrated that their revised DEB (rDEB) model outperformed the sDEB model for a measured time series of degradation of the herbicide 2,4-dichlorophenoxyacetic acid, and both DEB models resulted in better explanation than the popular compromise model for the empirically observed relationships (1) between specific respiration and specific growth rate for the marine bacterial data collected by Vikström and Wikner (2019), and (2) between specific substrate uptake rate and biomass yield for the microbial data synthesis by Smeaton and Van Cappellen (2018). However, in further exploration of the rDEB
model, Tang and Riley (2023) noticed (in their section 4.2) that the sDEB model is paradoxically recovered from the rDEB only when the ribosome effort allocated to growth is zero. Thus, in the following, we derive the reserve dynamics using an alternative approach that combines the first law of thermodynamics and law of mass action, which then leads to the mDEB model. As we show below, the mDEB model remains compatible with all key assumptions of the sDEB model, and can be easily extended to non-linear enzyme kinetics-based model of intracellular metabolism. Moreover, the mDEB model extension
to multiple-substrate-limited biological growth is computationally much more efficient, while maintaining the theoretical elegance of the DEB theory. We note that this gained computational efficiency by the mDEB model will be very helpful to apply the DEB theory to the modelling of a large number of concurrently growing organism (or organs for plants and animals).





Below we first provide a detailed derivation of the mDEB model. Then the mDEB model is compared to the sDEB model for general behaviours. Following this, an application to the chemostat experiment by Pawlowski (2004) is presented.

Finally, we conclude by discussing how the mDEB model can be applied to growth problems in ecosystem biogeochemistry.

## 2. Theory

### 2.1. The mDEB model

We start with the mass (or energy) conservation equation from the non-equilibrium thermodynamics (De Groor and

Mazur, 1984),

$$\frac{d}{dt}\int_{V(t)}\rho_i dV = -\int_{\boldsymbol{\Omega}(t)}\boldsymbol{J}_i \cdot d\boldsymbol{\Omega} + \int_{V(t)}\sum_l \sigma_{li} r_l(\rho_l)\, dV, \tag{1}$$

where subscript $i$ refers to pools internal to the time dependent volume $V(t)$; $\rho_i$ is the mass (or energy) density of the $i$-th internal pool that is enclosed by the volume $V(t)$, whose surface is $\boldsymbol{\Omega}(t)$ (and its normal direction is pointing outward); $\boldsymbol{J}_i$ is the outgoing flux of $\rho_i$ through the surface $\boldsymbol{\Omega}(t)$; and $r_l(\rho_l)$ is the $l$-th chemical reaction rate that occurs inside the volume $V(t)$ contributing to the change of total mass (or energy) $\int_{V(t)}\rho_i dV$ according to the stoichiometric parameter $\sigma_{li}$. We define

all the symbols in the Nomenclature table in the appendix, and unless specified otherwise, all variables have ISO units.

When applied to a biological cell, $V(t)$ is its physical volume, and $\boldsymbol{\Omega}(t)$ is the corresponding exterior surface. The $i$-th internal reserve is $\rho_i$, whose dynamics are governed by substrate uptake ($-\boldsymbol{J}_i$), and intracellular chemical transformations (by the last term of equation (1)). Since equation (1) imposes no size restriction on the spatial domain of the integral terms *a priori*, as the sDEB model has attempted to achieve, the mDEB model is applicable to unicellular, multicellular, and a

(sub)population of organisms, as long as some average is properly taken in the application. Moreover, as in the sDEB model, the volume $V(t)$ and surface $\boldsymbol{\Omega}(t)$ here are assumed to be scaled based on structural biomass (through mass density, which is constant by the strong homeostasis assumption).

When applying equation (1) to a reserve component of a biological organism or a population of cells, we ignore the location dependence within the intracellular volume. That is, $\rho_i$ is the average value within the volume $V(t)$. Applying Gauss's

law to the first term of equation (1) (e.g., Feynman et al., 2011) leads to

$$\frac{d}{dt}\left(\bar{\rho}_i(t)V(t)\right) = -\int_{V(t)}\boldsymbol{\nabla}\cdot\boldsymbol{J}_i dV + \sum_l \sigma_{li} r_{li}(\bar{\rho}_l)\, V(t), \tag{2}$$

where "$\bar{\square}$" signifies a spatial average for the variable. Equation (2) will be used to formulate the reserve dynamics for the mDEB model. Additionally, for simplicity, we henceforth remove the explicit designation of time dependence in $V(t)$.

In the following, we consider the simplest, one reserve mDEB model. However, in order to show that the mDEB model satisfies the merging and partitioning principle of the reserve dynamics, we partition the reserve (density) as

$$x = \sum_i x_i. \tag{3}$$





With equation (3), by taking $x_i = \bar{\rho}_i(t)$, equation (2) implies that, while all reserve molecules are chemically the same, in the model, they can be tagged with subscript $i$, so that one can differentiate them and model them separately. To some extent, this tagging is equivalent to isotope labelling, if the isotopes are not metabolically differentiated by the organism.

The application of equation (2) to $x_i$ leads to

$$\frac{d(Vx_i)}{dt} = j_{A,i}V - Vj_{E,x_i} = j_{A,i}V - V\frac{v_E x_i/K_i}{1+\sum_l x_l/K_l} = j_{A,i}V - V\frac{v_E x_i/K_i}{1+x/K}. \tag{4}$$

where $j_{A,i}$ is the specific assimilation rate from substrates contributing to $x_i$, and $v_E$ is the maximum specific reserve

mobilization rate (as signified by subscript $E$). Since all reserve compartments $x_i$ are metabolically the same, their specific affinities are also the same: $K_i = K_l = K$.

In equation (4), we have formulated the intracellular biochemical reaction (i.e., $j_{E,x_i}$ for the turnover of $x_i$) using the equilibrium chemistry approximation (ECA) kinetics (which is a first order approximation to law of mass action; Tang (2015); Tang and Riley (2013, 2017)) and ignored the size contrast effect between intracellular substrates and enzymes (Tang and

Riley, 2019).

By summing up all parts with equation (4), we then obtain

$$\frac{d(Vx)}{dt} = j_A V - j_{E,x}V = j_A V - \frac{v_E x/K}{1+x/K}V, \tag{5}$$

which (by the chain rule of differentiation) can be written as

$$\frac{dx}{dt} = j_A - j_{E,x} - \mu x = j_A - \frac{v_E x/K}{1+x/K} - \mu x, \tag{6}$$

which describes the reserve dynamics of the single-reserve mDEB model.

The specific growth rate $\mu$ is computed from the dynamic energy budget as

$$\mu = \frac{1}{V}\frac{dV}{dt} = Y_V j_{E,x} - m_V = Y_V\frac{v_E x/K}{1+x/K} - m_V, \tag{7}$$

where $Y_V$ is the mass yield of converting reserve into structural biomass when considering the coupling between catabolism and anabolism, and $m_V$ is the specific maintenance rate. Equation (7) suggests that when reserve mobilization is too low, growth rate will become negative. Further, if mortality from various causes is to be included, equation (7) can be modified by adding the specific mortality rate.

In short, the single-reserve mDEB model is formulated by equations (6) and (7). When needed, the $\kappa$-rule for

allocation to soma expense (section 2.4 in Kooijman (2009)) can be incorporated by multiplying $Y_V$ with $\kappa$ (see Table 1). Moreover, the compatibility with ECA kinetics suggests that the reserve dynamics can be replaced with nonlinear kinetics-based models of intracellular metabolism (Tadmor and Tlusty, 2008; Etienne et al., 2020), such that the link with flux-balance models can also be naturally established (which is one major motivation for developing the rDEB model).

**2.2 Growth under weak homeostasis**





Under weak homeostasis, as food density is constant, specific reserve assimilation from external substrates $j_A$ is constant, leading to $\frac{dx}{dt} = 0$. When these conditions are applied to equation (6), one obtains

$$j_{E,x} = \frac{v_E x}{K+x} = j_A - \mu x. \tag{8}$$

which, when substituted into the dynamic energy budget equation (7), leads to

$$\mu = Y_V(j_A - \mu x) - m_V, \tag{9}$$

such that

$$\mu = \frac{Y_V j_A - m_V}{1 + Y_V x}. \tag{10}$$

Equation (10) states that under constant food density, the mDEB model predicts that growth continues with constant reserve density, just as predicted by the sDEB theory and required by the observed weak homeostasis.

From equations (8) and (9), the reserve density $x$ can be found as

$$x = \frac{(\mu + m_V)K}{Y_V v_E - \mu - m_V}. \tag{11}$$

When equation (11) is entered into equation (9), one finds

$$j_A = \frac{\mu + m_V}{Y_V} + \mu x = \frac{\mu + m_V}{Y_V} \frac{Y_V v_E - m_V - \mu(1 - KY_V)}{Y_V v_E - m_V - \mu}. \tag{12}$$

Equation (12) can be used to derive the yield of structural biomass (or population) under weak homeostasis:

$$Y_\mu = \frac{\mu}{j_A} Y_X = \left(\frac{\mu}{\mu + m_V}\right)\left(\frac{Y_V v_E - m_V - \mu}{Y_V v_E - m_V - \mu + Y_V K \mu}\right) Y_V Y_X, \tag{13}$$

where $Y_X$ is the reserve biomass yield for assimilating substrate from the environment.

Accordingly, the yield for total biomass is

$$Y_B = Y_\mu(1 + x) = \left(\frac{\mu}{\mu + m_V}\right)\left(\frac{Y_V v_E + (m_V + \mu)(K-1)}{Y_V v_E - m_V - \mu + Y_V K \mu}\right) Y_V Y_X, \tag{14}$$

Since the mDEB model is compatible with the weak homeostasis assumption, like the sDEB model, it is naturally compatible with the Von Bertalanffy growth model that relates the size of an organism to its age at constant specific food supply (see section 2.6.1 in Kooijman (2009)). The Von Bertalanffy growth model states that, under constant food supply, an 130  organism grows exponentially over time to reach its maximum size. Moreover, under this condition, since $j_A$ is constant, the specific growth rate $\mu$ can be solved from the equation of $Y_\mu$ for the mDEB model in Table 1 (which can be verified to be a quadratic equation of $\mu$; see its special case in equation (21)).

## 2.3. mDEB model for $K \gg x$

Under the condition $K \gg x$ (i.e. the high enzyme condition that is usually satisfised inside biological cells; Tang and 135  Riley (2023); Phillips et al. (2012)), we may define $\tilde{v}_E = v_E/K$, so that equation (6) becomes

$$\frac{dx}{dt} = j_A - \tilde{v}_E x - \mu x, \tag{15}$$





and equation (7) becomes

$$\mu = Y_V \tilde{v}_E x - m_V. \tag{16}$$

For growth under weak homeostasis (aka constant specific food supply and therefore constant reserve density), one then has

$$x = \frac{\mu + m_V}{Y_V \tilde{v}_E}, \tag{17}$$

and

$$j_A = (\tilde{v}_E + \mu)x = \left(\frac{\tilde{v}_E + \mu}{\tilde{v}_E}\right)\left(\frac{\mu + m_V}{Y_V}\right), \tag{18}$$

and, accordingly, the structural biomass yield is

$$Y_\mu = \frac{\mu}{j_A} Y_X = \left(\frac{\mu}{\mu + m_V}\right)\left(\frac{\tilde{v}_E}{\tilde{v}_E + \mu}\right) Y_V Y_X, \tag{19}$$

and the total biomass yield is

$$Y_B = Y_\mu(1 + x) = \left(\frac{\mu}{\mu + m_V}\right)\left(\frac{Y_V \tilde{v}_E + \mu + m_V}{\tilde{v}_E + \mu}\right) Y_X. \tag{20}$$

Additionally, from equation (18), one can obtain

$$\mu = \frac{m_V + \tilde{v}_E}{2}\left(-1 + \sqrt{1 + \frac{4\tilde{v}_E(j_A Y_V - m_V)}{(m_V + \tilde{v}_E)^2}}\right). \tag{21}$$

## 3. Comparisons with the sDEB model

### 3.1 Biological growth on single reserve.

The mDEB model is compared to the sDEB model for growth from a single reserve pool in Table 1. That comparison shows that the two models have the same equation for specific growth rate ($\mu$ vs $\mu_s$) only under weak homeostasis (when the reserve density reaches steady state and thus the whole organism is of fixed elemental stoichiometry), and, mathematically, the two models are structurally very similar.

Under the weak homeostasis condition, because reserve density is time invariant and depends algebraically on reserve assimilate rate ($j_A$), the mDEB and sDEB models predict the specific structural biomass growth rate as a function of substrate concentration in a pattern very similar to predicted using Monod kinetics (Figure 1; Monod (1949)). When the specific reserve turnover rate is much greater than the specific maintenance rate, the mDEB model for $K \gg x$ and the sDEB model predict almost identical growth rates as a function of substrate availability (lines with $v_E = 100 m_V$ in Figure 1).

Moreover, the two models predict very similar relationships between specific growth rate and both structural biomass yield and total biomass yield under weak homeostasis (Figure 2). Specifically, under weak homeostasis conditions, both the mDEB and sDEB models predict that the structural biomass yield first increases, then plateaus, and then decreases with specific growth rate (Figure 2a, c), with the sDEB model predicting a faster decrease at higher growth rate. The total biomass yield





increases almost hyperbolically with specific growth rate (Figure 2b, d), with the sDEB model predicting a faster increase at

higher growth rate. We note that the relationship between total biomass yield and specific growth rate is consistent with that predicted by the empirical compromise model (Beeftink et al., 1990; Wang and Post, 2012). The qualitative difference between structural biomass yield and total biomass yield suggests that more analysis is needed on how they may affect simulated biogeochemistry.

Table 1. Mathematical comparison of the mDEB and the sDEB models, with sDEB symbols annotated by subscript "s". For both models, the $\kappa$-rule for allocation to soma is applied (Kooijman, 2009). Also, it is assumed that structural biomass is proportional to the cellular population. Additionally, $j_A$ and $j_{A,s}$ have incorporated the reserve yield $Y_X$ from substrate assimilation.

| mDEB model | sDEB model |
|---|---|
| Reserve dynamics: | Reserve dynamics: |
| $\frac{dx}{dt} = j_A - \frac{v_E x/K}{1+x/K} - \mu x$ | $\frac{dx_s}{dt} = j_{A,s} - v_{E,s} x_s$ |
| Structural biomass dynamics: | Structural biomass dynamics: |
| $\frac{dV}{dt} = \mu V$ | $\frac{dV_s}{dt} = \mu_s V_s$ |
| Specific structural biomass growth rate: | Specific structural biomass growth rate: |
| $\mu = Y_V \kappa \frac{v_E x/K}{1+x/K} - m_V$ | $\mu_s = \frac{\kappa_s Y_{V,s} v_{E,s} x_s - m_{V,s}}{1 + \kappa_s Y_{V,s} X_s}$ |
| Weak homeostasis condition | |
| Specific growth rate: | Specific growth rate: |
| $\mu = \frac{Y_V \kappa j_A - m_V}{1 + Y_V \kappa x}$ | $\mu_s = \frac{Y_{V,s} \kappa_s j_{A,s} - m_{V,s}}{1 + Y_{V,s} \kappa_s x_s}$ |
| Structural biomass yield vs growth rate: | Structural biomass yield vs growth rate: |
| $Y_\mu = \frac{\mu}{j_A} Y_X = \left(\frac{\mu}{\mu+m_V}\right)\left(\frac{\kappa Y_V v_E - m_V - \mu}{\kappa Y_V v_E - m_V + \mu(\kappa Y_V K - 1)}\right)\kappa Y_V Y_X$ | $Y_{\mu,s} = \frac{\mu_s}{j_{A,s}} Y_X = \left(\frac{\mu_s}{\mu_s+m_{V,s}}\right)\left(1 - \frac{\mu_s}{v_{E,s}}\right)\kappa_s Y_{V,s} Y_X$ |
| Total biomass yield vs growth rate: | Total biomass yield vs growth rate: |
| $Y_B = Y_\mu(1+x) = \left(\frac{\mu}{\mu+m_V}\right)\left(\frac{\kappa Y_V v_E + (m_V+\mu)(K-1)}{\kappa Y_V v_E - m_V - \mu(1-\kappa Y_V K)}\right)\kappa Y_V Y_X$ | $Y_{B,s} = Y_{\mu,s}(1+x_s) =$ $\left(\frac{\mu_s}{\mu_s+m_{V,s}}\right)\left(\frac{\kappa_s Y_{V,s} v_{E,s} + m_{V,s} + (1-\kappa_s Y_{V,s})\mu_s}{v_{E,s}}\right) Y_X$ |








Figure 1. Comparison of predicted specific structural biomass growth rate as a function of normalized substrate availability
under weak homeostasis conditions. For cases with $v_E = 10m_V$, specific reserve turnover rate is 10 times the specific structural
biomass maintenance rate. For cases with $v_E = 100m_V$, specific reserve turnover rate is 100 times the specific structural
biomass maintenance rate. The mDEB model is based on equation (21), while the sDEB model is based on $\mu_s = \frac{Y_{V,s}j_{A,s}-m_{V,s}}{1+Y_{V,s}j_{A,s}/v_{E,s}}$.
In producing the above results, for both $j_A$ and $j_{A,s}$, the Michaelis-Menten kinetics $f(S) = j_{A,max} S/(S + K_S)$ is used for
substrate uptake.



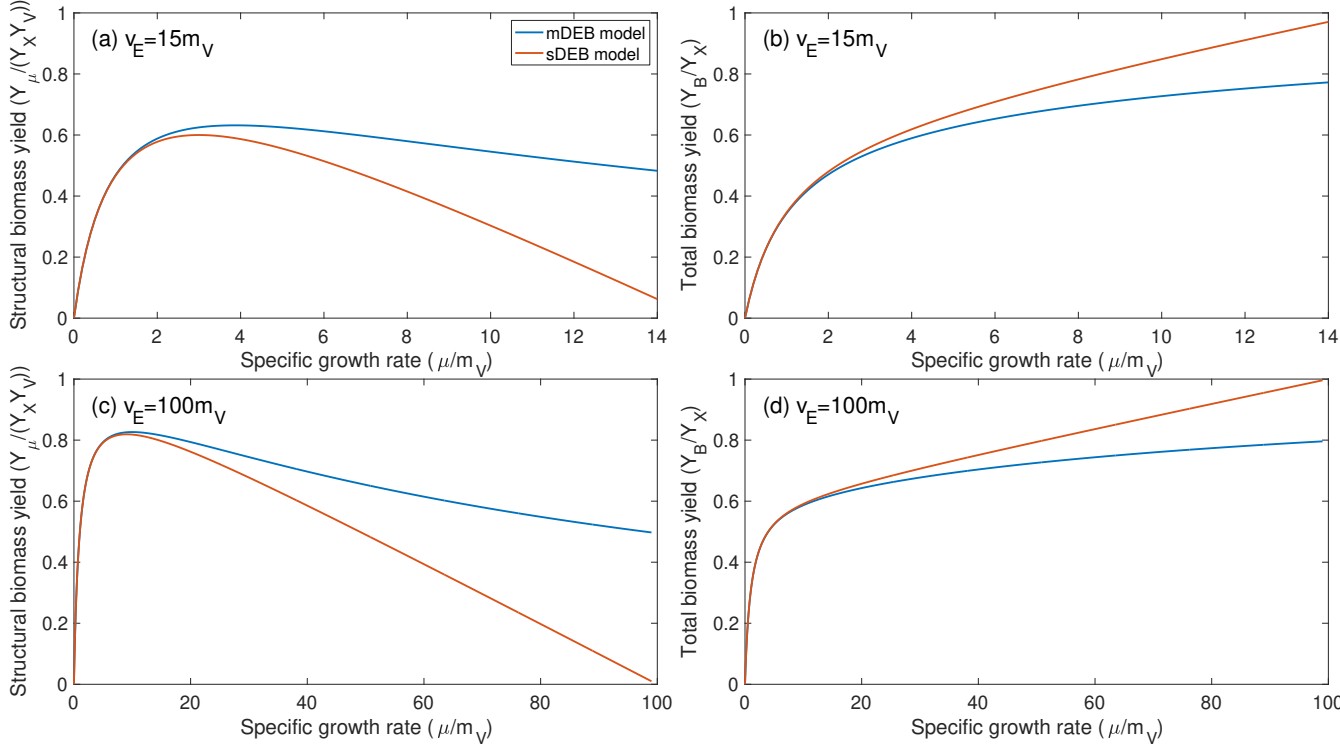

Figure 2. Comparison of predicted biomass yield as a function of normalized specific growth rate under weak homeostasis conditions. Left panels (a) and (c): predicted relationship between structural biomass yield and specific growth rate. Right panels (b) and (d): predicted relationship between total biomass yield and specific growth rate. $Y_X$ is the reserve biomass yield for substrate assimilation from the environment. For both models, it is assumed that the structural biomass yield from reserve biomass is 0.6. For upper panels (a) and (b), the maximum specific reserve turnover rate is 15 times the specific structural biomass maintenance rate. For lower panels (c) and (d), the maximum specific reserve turnover rate is 100 times the specific structural biomass maintenance rate.

### 3.2 Biological growth on two reserves

When an organism is growing on two complementary reserves (from assimilation two complementary substrates, e.g., carbon and nitrogen), the growth rate may be computed using the synthesizing unit kinetics (Kooijman, 2009), such that

$$\mu = \left( j_{G,x_1}^{-1} + j_{G,x_2}^{-1} - \left( j_{G,x_1} + j_{G,x_2} \right)^{-1} \right)^{-1}. \tag{22}$$

For simplicity, in equation (22) we assumed that the specific growth rate $\mu$ is positive. Negative growth when reserve fluxes fall short of maintenance requirements can be considered separately (Tolla et al., 2007; Kooijman, 2009).

By designating the maintenance flux required for reserve $x_1$ and $x_2$ as $j_{M,x_1}$ and $j_{M,x_2}$, one has

$$j_{G,x_i} = y_{G,x_i} \left( j_{E,x_i} - j_{M,x_i} \right), i = 1,2, \tag{23}$$

where $y_{G,x_i}$ is the stoichiometric yield of transforming reserve $x_i$ into structural biomass.





195        When equations (22) and (23) are applied with the mDEB model, according to equation (4), $j_{E,x_i}$ is independent of specific growth rate $\mu$, so that $\mu$ is an explicit function of fluxes $j_{E,x_i}$ and $j_{M,x_i}$. In contrast, when the sDEB model is applied, $j_{E,x_i}$ is a function of $\mu$, that is

$$j_{E,x_i} = (v_E - \mu)x_i. \tag{24}$$

Therefore, for the sDEB model, equations (22)-(24) indicate that $\mu$ is an implicit function, whose solution requires numerical iteration for each calculation of growth rate $\mu$ (also see equation of $\mu_g$ for the sDEB model in

200        Table 2). Furthermore, as the sDEB model strives to include more organisms and more complementary reserves, e.g. biological growth that is concurrently regulated by pools of carbon, nitrogen, and phosphorus (as many existing biogeochemical models attempt; e.g., Goll et al. (2012), Yu et al. (2020); Zhu et al. (2019), Mekonnen et al. (2019)), the necessity of iteration to explicitly represent many numbers of biological organisms will make the growth rate in the sDEB model increasingly more cumbersome to solve. In contrast, the absence of numerical iteration in the mDEB model will

significantly simplify this aspect of the modelling processes.

To demonstrate the applicability of mDEB model on biological growth over multiple complementary substrates, we constructed a mDEB model for the chemostat experiment of the diatom *Thalassiosira weissflogii* (*T. weissflogii*) from Pawlowski (2004). Like the sDEB model by Lorena et al. (2010), the mDEB model (

       Table 2) makes the following assumptions: (1) when the mobilized reserve fluxes fall short of the demand from

maintenance, structural biomass is reduced proportional to the deficit (as the maximum deficit of two reserves), and this reduced structural biomass is mineralized immediately into substrates; and (2) the fraction of rejected reserve that is not returned to reserve is not mineralized to become substrates. The mDEB model adopts almost identical parameter values from the sDEB model (compare our Table 3 with their Table 2), except that the mDEB model sets the specific reserve turnover rate ($v_E$) to 0.55 d⁻¹, calculated from manual tuning, while the sDEB model in Lorena et al. (2010) used a value of 2.60 d⁻¹. For

comparison, we also coded a sDEB model that differs from the mDEB model only in the computation of growth rate, which is achieved through the bisection algorithm (Burden and Faires, 1985). Because we adopted a formulation of negative growth rate $\mu_d$ different from Lorena et al. (2010) (see their equation (2.11), where they assumed that negative growth rates from carbon reserve deficit and nitrogen reserve deficit are additive. However, we used the maximum of the two instead), the sDEB model here used 0.95 d⁻¹ for the specific reserve turnover rate.

220        By using parameters mostly from the literature, we find that the mDEB model predictions captured the measured time series of particulate organic carbon and nitrogen reasonably (Figure 3). While better model-data agreement may be obtained by optimizing more parameters, the results here suggest that the mDEB model is at least as capable as the sDEB model. Interestingly, the sDEB model here produced almost identical model-data agreement. Moreover, for the 30-day simulation period, the mDEB model took about 55% of the time of the sDEB model. On an Apple M3 Max machine with 64 GB memory,

using MatlabR2020b, the typical execution time are 0.037 seconds and 0.068 seconds for the mDEB model and sDEB model,



respectively. (We provide the source code for readers to play with the models.) This significant improvement in computational efficiency and good model-data agreement thus suggests mDEB model is a good replacement of the sDEB model.

Table 2. A mDEB model of diatom *T. weissflogii* growing on $CO_2$ and inorganic nitrogen. For comparison, also given are sDEB model equations for growth and reserve dynamics.

| mDEB model equations | Description |
|---|---|
| $j_{A,N} = j_{Am,N}\frac{[N]}{[N]+K_N}$ | Specific assimilation rate of substrate N; molN $(\mathrm{mol}M_V)^{-1}$ d$^{-1}$. |
| $j_{\mathrm{ph}} = \frac{\rho_{PSU}I}{1/\alpha+1/\gamma(1+\beta/\alpha)I+(\beta/\gamma\delta)I^2}$ | Photosynthesis. |
| $j_{CO_2} = j_{Am,CO_2}\frac{[CO_2]}{[CO_2]+K_C}$ | Specific $CO_2$ flux rate for C-reserve synthesis; molC $(\mathrm{mol}M_V)^{-1}$ d$^{-1}$. |
| $j_{A,C} = \left(\frac{1}{j_{CO_2}} + \frac{1}{j_{\mathrm{ph}}} - \frac{1}{j_{\mathrm{ph}}+j_{CO_2}}\right)^{-1}$ | Specific carbon-reserve synthesis rate from $CO_2$; molC $(\mathrm{mol}\ M_V)^{-1}$ d$^{-1}$. |
| $\frac{dx_C}{dt} = j_{A,C} - (v_E + \mu)x_C + \kappa_E j_{R,C}$ | Dynamic equation of C-reserve density. |
| $\frac{dx_N}{dt} = j_{A,N} - (v_E + \mu)x_N + \kappa_E j_{R,N}$ | Dynamic equation of N-reserve density. |
| $j_{G,i} = v_E x_i - j_{M,i}, i = C$ or $N$ | Reserve fluxes to support potential structural biomass growth. |
| $j_{R,i} = j_{G,i} - y_{V,i}\mu_g, i = C$ or $N$ | Potentially rejected reserve flux in intracellular metabolism. |
| $\mu_g = max\left[\left(\left(\frac{j_{G,C}}{y_{V,C}}\right)^{-1} + \left(\frac{j_{G,N}}{y_{V,N}}\right)^{-1} - \left(\frac{j_{G,C}}{y_{V,C}} + \frac{j_{G,N}}{y_{V,N}}\right)^{-1}\right)^{-1}, 0\right]$ | Synthesizing unit kinetics based structural growth rate. |
| $\mu_d = max\left(-\frac{j_{G,C}}{y_{V,C}}, -\frac{j_{G,N}}{y_{V,N}}, 0\right)$ | Structural biomass respired due to maintenance deficit. |
| $\mu = \mu_g - \mu_d$ | Net specific growth rate of structural biomass. |
| $j_{X,i} = min(v_E x_i, j_{M,i}) + (y_{V,i} - n_{V,i})\mu_g + n_{V,i}\mu_d, \ i = C$ or $N$ | Respiration flux as $CO_2$ and inorganic N added to substrate pools. |
| $\frac{dM_V}{dt} = (\mu - h)M_V$ | Dynamic equation of structural biomass. |
| $\frac{d[CO_2]}{dt} = h([CO_2]_r - [CO_2]) - (j_{A,C} - j_{X,C})M_V$ | Dynamic energy of dissolved $CO_2$. |
| $\frac{d[N]}{dt} = h([N]_r - [N]) - (j_{A,N} - j_{X,N})M_V$ | Dynamic energy of dissolved inorganic nitrogen. |

| sDEB model equations | |
|---|---|
| $\frac{dx_C}{dt} = j_{A,C} - v_E x_C + \kappa_E j_{R,C}$ | Dynamic equation of C-reserve density. |
| $\frac{dx_N}{dt} = j_{A,N} - v_E x_N + \kappa_E j_{R,N}$ | Dynamic equation of N-reserve density. |
| $j_{G,i} = (v_E - \mu)x_i - j_{M,i}, i = C$ or $N$ | Reserve fluxes to support potential structural biomass growth. |



$$\mu_g = \left( \left( \frac{j_{G,C}(\mu_g)}{y_{V,C}} \right)^{-1} + \left( \frac{j_{G,N}(\mu_g)}{y_{V,N}} \right)^{-1} - \left( \frac{j_{G,C}(\mu_g)}{y_{V,C}} + \frac{j_{G,N}(\mu_g)}{y_{V,N}} \right)^{-1} \right)^{-1}$$

Synthesizing unit kinetics based structural growth rate.

Table 3. Parameters of the mDEB model. The values of $v_E$ were found by manual tunning, with the sDEB model value in the braces. All other parameters are the same as in Lorena et al. (2010).

| Parameter | Description | Units | Value | Reference |
|---|---|---|---|---|
| $n_{V,N}$ | N to C ratio of structural biomass | molN (molC)$^{-1}$ | 0.04 | Geider et al. (1998); Baklouti et al. (2006) |
| $j_{Am,N}$ | Maximum volume-specific N assimilation | molN (molM$_V$)$^{-1}$ d$^{-1}$ | 1.0 | Geider et al. (1998) |
| $j_{Am,CO_2}$ | Maximum volume-specific $CO_2$ assimilation | mol C (molM$_V$)$^{-1}$ d$^{-1}$ | 5.1 | Geider et al. (1998) |
| $K_C$ | Half-saturation concentration for $CO_2$ uptake | $\mu$M | 0.43 | Rost et al. (2003) |
| $K_N$ | Half-saturation concentration for N uptake | $\mu$M | 3.20 | Pawlowski et al. (2002) |
| $j_{M,C}$ | Volume-specific maintenance cost paid by C reserve | molEC (molM$_V$)$^{-1}$ d$^{-1}$ | 0.054 | Faugeras et al. (2004); Quigg and Beardall (2003) |
| $j_{M,N}$ | Volume-specific maintenance cost paid by N reserve | molEN (molM$_V$)$^{-1}$ d$^{-1}$ | 0.012 | Faugeras et al. (2004); Quigg and Beardall (2003) |
| $v_E$ | Specific reserve turnover rate | d$^{-1}$ | 0.55(0.95) | Calibrated |
| $y_{V,C}$ | Yield factor of C-reserve to structure | molEC (molM$_V$)$^{-1}$ | 1.25 | Baklouti et al. (2006) |
| $y_{V,N}$ | Yield factor of N-reserve to structure | molEN (molM$_V$)$^{-1}$ | 0.04 | Lorena et al. (2010) |
| $\kappa_E$ | Fraction of rejection flux incorporated into C or N-reserve | — | 0.7 | Lorena et al. (2010) |
| $\alpha$ | PSU excitement coefficient | (mmolPSU $\mu$Em$^{-2}$)$^{-1}$ | 0.0019 | Wu and Merchuk (2001) |
| $\beta$ | PSU inhibition coefficient | (mmolPSU $\mu$Em$^{-2}$)$^{-1}$ | 5.8E − 7 | Wu and Merchuk (2001) |
| $\gamma$ | PSU relaxation rate | mmolPSU$^{-1}$ s$^{-1}$ | 0.1460 | Wu and Merchuk (2001) |
| $\delta$ | PSU recovery rate | mmolPSU$^{-1}$ s$^{-1}$ | 4.8E − 4 | Wu and Merchuk (2001) |
| $\rho_{PSU}$ | PSU density | mmolPSU (molM$_V$)$^{-1}$ | 0.365 | Wu and Merchuk (2001) |





Figure 3. Model data comparison for the chemostat experiment of *T. weissflogii* from Pawlowski (2004). (a) Photon flux density (PFD) over the measurement period; (b) particulate organic carbon (reserve plus structural biomass carbon; POC) of the *T. weissflogii* population; (c) particulate organic nitrogen (reserve plus structural biomass nitrogen; PON) of the *T. weissflogii* population. All panels only show results over the time period when measurements were available.





## 4. Conclusions

240         Starting with the mass conservation equation from the nonequilbrium thermodynamics, law of mass action and the basic assumptions of the dynamic energy budget theory, we derived a modified formulation of the dynamic energy budget (mDEB) model. The mDEB model is mathematically very similar to the standard dynamic energy budget (sDEB) model and is able to recover many of the features of the sDEB model. However, because it does not require numerical iterations for the

computation of growth rate, particularly for biological growth that involves multiple complementary substrates, the mDEB model is computationally much more efficient. In the example application to a chemostat experiment of *T. weissflogii,* the mDEB and sDEB models are found equally accurate, while the former only required almost half the computing time of the latter. Moreover, since the mDEB model is compatible with non-linear kinetics for reserve turnover, it can be extended into models that consider ribosome allocation explicitly for microbes (Tadmor and Tlusty, 2008).

250         With its strong theoretical foundation and easier numerical implementation, the mDEB model consistently formulates biological growth of microbes, plants, and animals (which are all in the application domain of the DEB theory; Kooijman (2009); Yang et al. (2020); Russo et al. (2022); Matyja and Lech (2024)). Its consistent application to ecosystem biogeochemical models will help modelers alleviate structural uncertainty, as advocated in Tang et al. (2024).

## Appendix A: Nomenclature


Below only includes symbols not defined in Table 2 and Table 3.

| Symbol | Unit | Description |
|---|---|---|
| $j_{A,i}, j_A$ | s$^{-1}$ | Specific reserve assimilate rate. |
| $j_{E,x}$ | s$^{-1}$ | Specific reserve turnover rate. |
| $m_V$ | s$^{-1}$ | Specific structural biomass maintenance. |
| $r_l$ | kg m$^{-3}$ s$^{-1}$ | Rate of *l*-th reaction. |
| $\tilde{v}_E$ | s$^{-1}$ | Specific reserve turnover rate. |
| $x, x_i$ | kg reserve (kg structure)$^{-1}$ | Reserve density. |
| $\boldsymbol{J}_i$ | kg m$^{-2}$ | Mass flux density. |
| $K, K_l$ | kg reserve (kg structure)$^{-1}$ | Half saturation parameter |
| $V(t), V$ | m$^{-3}$ (or kg m$^{-3}$) | Volume (or structural biomass). |
| $Y_V$ | kg structure (kg reserve)$^{-1}$ | Structural biomass yield from reserve biomass. |
| $Y_\mu$ | kg structure (kg substrate)$^{-1}$ | Emergent structural biomass yield from substrate assimilation. |
| $Y_B$ | kg biomass (kg substrate)$^{-1}$ | Emergent total biomass yield from substrate assimilation. |
| $\rho_i, \rho_l$ | kg m$^{-3}$ | Mass density. |



| | | |
|---|---|---|
| $\bar{\rho}_i, \bar{\rho}_l$ | kg m$^{-3}$ | Space-averaged mass density. |
| $\sigma_{li}$ | kg kg$^{-1}$ | Stoichiometry coefficient for substrate $i$ due to reaction $l$. |
| $\mu$ | s$^{-1}$ | Specific growth rate. |
| $\kappa$ | kg kg$^{-1}$ | Fraction of reserve turnover for soma development. |
| $\mathbf{\Omega}(t)$ | m$^2$ | Surface area of volume $V(t)$. |

**Author contributions.**

JT designed the study, conducted the analysis, and wrote the paper. WJR discussed the results and edited the paper.

**Competing interests.**

The contact author has declared that neither of the authors has any competing interests.

**Disclaimer.**

Financial support does not constitute an endorsement by the Department of Energy and National Science Foundation of the views expressed in this study.

**Financial support**.

This research has been supported by the director of the Office of Science, Office of Biological and Environmental Research, of the US Department of Energy (contract no. DE-AC02-05CH11231) as part of the Belowground Biogeochemistry Science Focus Area, the Reducing Uncertainties in Biogeochemical Interactions through Synthesis and Computation (RUBISCO) Scientific Focus Area. Jinyun Tang is also supported by the National Science Foundation (award no.2125069).

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
