# Peer review of "Technical Note: A modified formulation of dynamic energy budget theory for faster computation of biological growth"

_EGUsphere, 2024_

## Author Response (AR1)

**Response letter**

We appreciate the comments from two reviewers, which significantly helped us improve the readability of our manuscript.

Below, we repeat the reviewers' comments and then respond to each comment in blue. Related modifications in the revised manuscript are highlighted in red.

**Response to Reviewer 1**

**Comment 1**: This MS represents a useful contribution to bioenergetic modeling, beyond its stated emphasis as a technical note focused on accelerating computations of biological growth. This is because an overwhelmingly large proportion of the literature on DEB models is constructed around the assumptions of Kooijman's "standard" DEB model – called sDEB in the paper. One of the author's previous publications (Tang and Riley 2023) is an exception, an interesting reformulation (rDEB) that offers (and tests) a representation of reserve homeostasis with a more transparent link to known subcellular processes. The new model (mDEB) in this paper is a further variant that reduces in a limiting case to the sDEB model. The main emphasis, computational speed is of course important for two reasons: (i) population simulations and (ii) parameter estimation using computationally intensive methods. The latter is important as DEB models typically use state variables that are not directly observable.

**Response**: We appreciate the reviewer for highlighting the merit of our past and current manuscript. We hope our contribution to the DEB theory will help make it more popular in the research community, and eventually establish a unified framework for modeling all kinds of biological organisms under realistic conditions.

**Comment 2**: The reasoning behind the mDEB model is presented clearly. However, if I understand it correctly, the "high enzyme condition" (line 134) is essential for the numerical improvements proposed to be valid for systems with multiple reserves. Equation (22) for synthesizing unit kinetics approximates the form in Kooijmans work in the limiting situation where there is no upper limit to the reaction rate of the final step (release of product). If this is included calculation of the growth rate (mu) may well require handling a set of implicit equations and there can be situations in more complex models where these do not have a unique solution (see Pfab et al 2022 - 10.1093/conphys/coac026 for an example). THis point does not require changes in the main text, but, if correct, might merit mention in the discussion.

**Response**: We believe that there is some confusion in our presentation that triggered the reviewer's concern that our modification of DEB theory is limited to high enzyme conditions. We should have made it clear that focusing on "high enzyme concentrations" is only meant to

simplify our presentation. In practice, we note that, in the mDEB framework, the flux $j_{G,i}$ used to apply Equation (22) has no dependence on growth rate under all enzyme concentrations, which is noted in the last term of equation (4) and also in the formulation of $j_{G,i}$ for the mDEB model in Table 2. Therefore, in the revised manuscript, we now made it clear why we emphasize the high enzyme concentration case, and also highlight that the mDEB model does not require iteration under all intracellular enzyme concentrations. Specifically, in section 2.3, we now write "*We next apply the condition $K \gg x$ to derive some special case analytical results of the mDEB model. (We note that the condition $K \gg x$ corresponds to the high enzyme condition that is usually satisfised inside biological cells (Tang and Riley, 2023; Phillips et al., 2012), even though it is not essential for the application of the mDEB model.)*"

**Comment 3**: Derivation of the mDEB equations via equation (2) and Gauss's theorem will intimidate some readers. Those who do understand eq (2) may worry about the first term of the RHS (divergence of J). This is zero everywhere in the interior of the cell (by assumption), so at minimum the text needs to state that the integral is over the volume (including the surface).

**Response**: Thanks for pointing out this "intimidating factor". We now added more detailed explanations, so math lovers can enjoy the equations, and others can better understand our logic through verbal explanations. In particular, we added "*In plain language, equation **Error! Reference source not found.** states that the changing rate of total mass $\int_{V(t)} \rho_i dV$ within a volume of space $V(t)$ (i.e., left hand side) is determined by the mass flux escaping through its surface $\Omega(t)$ (i.e., the first term on the right hand side), and the net chemical production inside the volume (the second term in the right hand side).*"

**Comment 4**: In line 121, there is mention of "observed weak homeostasis". I doubt this; weak homeostasis is a central assumption of sDEB, but I know of no examples where an organism's composition remains constant through its lifetime.

**Response**: We revised the wording to better reflect the assumption made in the DEB theory.

**Comment 5**: The sentence beginning line129 "the Von Bertalanffy….." is not strictly accurate.

**Response**: We removed this sentence to avoid confusion.

**Response to reviewer 2**

**Comment 1**: This technical note introduces a modified formulation of the dynamic energy budget model that, while similar in structure to the standard dynamic energy budget model, is significantly more efficient computationally as it does not require an implicit calculation for the calculation of the growth rate. This may represent an important advantage when the model is applied to study the dynamics of multiple organisms and substrates. The manuscript is also well

written, the theory is described well, and the application and comparison with the standard model are fair. I recommend publication of this note, after considering the comment below.

**Response**: We appreciate the reviewer for recognizing the value of our work. In the following, we carefully addressed concerns you raised, hoping our paper is now better serving interested readers.

**Comment 2**: The key contribution of this manuscript is the introduction of a dynamic energy budget model, which is derived from the standard model. However, except for a quick summary in the abstract (lines 10-11), the manuscript does not detail how the new model differs from the previous formulation. It would be helpful if section 2 would include regular comparisons between what is being introduced here and what is done in the standard model. I do not think the section should be expanded much, but when needed there should be short sentences referring to the formulation in the standard model. This way, when applying the model to the case of two reserves, it would be clearer why $\mu$ is not an implicit function here, while it is in the standard formulation.

**Response**: Thanks for pointing out potential confusion here. We have added words to highlight the differences between our approach and the standard formulation.

**Comment 3**: I also noticed that the introduction, at the end, mentions that there would be a discussion of how the mDEB model could be applied to ecosystem biogeochemistry. However, I did not find such a discussion after the comparison with data. I am not sure if the authors are referring to the conclusion section, but here there is only one sentence stating that the application of the model would alleviate structural uncertainty. If the authors are referring to this sentence, I would elaborate a bit on this, for otherwise it is not really clear how and why the uncertainty would be alleviated.

**Response**: We now added more details on how the mDEB model could be applied to ecosystem biogeochemistry. Specifically, we added "*Last, we highlight that $j_{E,x}$ in the mDEB model (see equation (6)) has no explicit dependence on growth rate $\mu$. In contrast, in the sDEB model, based on a pure mathematical argument (see section 2.3.1 in (Kooijman, 2009)), $j_{E,x} + \mu x$ is formulated as a linear function of reserve density $x$, such that $j_{E,x}$ depends linearly on growth rate $\mu$. This difference offers mDEB significant computational advantage for modelling biological organisms growing on multiple complimentary nutrients, such as carbon, nitrogen, phosphorus, sulfur and even silicon (Madigan et al., 2009).*"

**Comment 4**: Equation 1 is a conservation of mass, not energy. The latter would require a distinction between heat and work, that goes beyond what is being done here. As for the citation, the book is by the *de Groot and Mazur*. See typo.

**Response**: We revised the text to clarify that equation (1) is about mass conservation, and removed the mentioning of energy. We also corrected the typo.

**Comment 5:** Line 72: the sentence "(and its normal direction is pointing outward)" may be confusing and perhaps not needed. I believe the authors are referring to the unit surface vector (needed to compute the surface integral), which by convention is normal to the surface and pointing outward.

**Response**: We removed this confusing comment.

**Comment 6**: Line 80: "as long as some average is properly taken in the application.". Equation 1 applies to any defined control volume (and control surface) and the integral of ρ over the volume would give the total mass inside the volume. Why would using the average be a requirement?

**Response**: For Line 80, we now add one example-based explanation: "*For example, for a population of individuals, even though the individuals may have different reserve density, the population reserve dynamics is represented using mean reserve density computed as the ratio between whole population reserve biomass and whole population structural biomass*".

**Other minor comments**:

Line 60: substrate.

Line 85: "first term" on the right-hand side.

Line 110: some

In Table 1, it may be useful to explicitly write the mathematical condition for weak homeostasis next to it. For example, Weak homeostasis condition (dx/dt=0, Ja = const).

 Line 218: I would not use the period inside a parenthesis. You could use a semi-column before "however".

Line 227: instead of "reader to play with the model", I would use "readers to test or adopt the model", or similar.

**Response**: We addressed all these editorial issues.